# The Roles of NOD-like Receptors in Innate Immunity in Otitis Media

**DOI:** 10.3390/ijms23042350

**Published:** 2022-02-21

**Authors:** Myung-Won You, Dokyoung Kim, Eun-Hye Lee, Dong-Choon Park, Jae-Min Lee, Dae-Woong Kang, Sang-Hoon Kim, Seung-Geun Yeo

**Affiliations:** 1Department of Radiology, College of Medicine, Kyung Hee University, Seoul 02447, Korea; funfun2020@khu.ac.kr; 2Department of Anatomy and Neurobiology, College of Medicine, Kyung Hee University, Seoul 02447, Korea; dkim@khu.ac.kr; 3Department of Pediatrics, College of Medicine, Kyung Hee University, Seoul 02447, Korea; leeeh80@gmail.com; 4St. Vincent’s Hospital, The Catholic University of Korea, Suwon 16247, Korea; dcpark@catholic.ac.kr; 5Department of Otorhinolaryngology—Head and Neck Surgery, School of Medicine, Kyung Hee University, Seoul 02447, Korea; sunjaesa@naver.com (J.-M.L.); kkang814@naver.com (D.-W.K.); hoon0700@naver.com (S.-H.K.)

**Keywords:** otitis media, effusion, granulation tissue, cholesteatoma, NOD-like receptor

## Abstract

Acute otitis media (AOM) can persist or lead to various complications in individuals in which the innate immune system is impaired. In this context, impaired expression of nucleotide-binding oligomerization domain (NOD)-like receptor (NLR), an intracellular pathogen-recognition receptor (PRR), is involved in the etiology of OM in humans and animals, affecting its development, severity, chronicity, recurrence, and associated complications. To assess this relationship, we reviewed literature reports relating NLR expression patterns with the pathophysiology and clinical features of OM in the larger context of impaired innate immunity. We summarized the results of published studies on the expression of NLRs in animals and humans in acute otitis media (AOM), otitis media with effusion (OME), chronic otitis media (COM) with cholesteatoma, and COM without cholesteatoma. NLRs were expressed mainly in association with bacterial infection in AOM, OME, COM with cholesteatoma, and COM without cholesteatoma. In addition, expression of NLRs was affected by the presence or absence of bacteria, fluid characteristics, disease recurrence, tissue type, and repeated surgery. Various factors of the innate immune system are involved in the pathogenesis of OM in the middle ear. NLRs are expressed in AOM, OME, COM with cholesteatoma, and COM without cholesteatoma. Impaired NLR expression induced the development, chronicity and recurrence of OM and exacerbated associated complications, indicating that NLRs have important roles in the pathogenesis of OM.

To investigate the expression and role of NLRs in OM, previously published studies of the associations of NLR expression patterns with the pathophysiology and clinical features of OM were identified. Literature databases were searched for studies on NLRs published in English or Korean. Studies were included if they (1) were prospective investigational studies; and (2) were studies in animals or human patients diagnosed with OM, AOM, OME, COM without cholesteatoma, or COM with cholesteatoma. Abstracts, case reports, and unpublished data were excluded. Studies published on or before 1 October 2021, were retrieved from four electronic databases: (1) SCOPUS (2) PubMed, (3) the Cochrane Library, and (4) EMBASE. Keywords included otitis media and NOD-like receptor. Ultimately, 11 articles satisfying the search criteria were selected for comprehensive review.

## 1. Overview of Otitis Media

Among the pathologies in the field of otolaryngology that have not yet been completely resolved is otitis media (OM), an otologic disease of infants and children. Acute otitis media (AOM) is an acute inflammation of the middle ear cavity that lasts ~3 weeks from the time of onset. Otitis media with effusion (OME) is diagnosed in cases where there is no perforation in the tympanic membrane and inflammatory fluid accumulates in the middle ear cavity. In cases where there is a perforation in the tympanic membrane in association with purulent discharge, the condition is called suppurative otitis media. Chronic otitis media (COM) is subdivided into COM with cholesteatomata and COM without cholesteatoma. Most AOM cases heal without sequelae, but in some cases, inflammation recurs or continues, leading to the development of recurrent OM or OME. In addition, if the inflammation of the middle ear cavity is not completely treated afterward, it may develop into a form of COM (Figure 1) [1,2,3].

The causes of AOM are very diverse and include viral or bacterial infection; eustachian tube dysfunction; allergy; anatomical, physiological, pathological and/or immunological factors in the middle ear; and environmental and genetic factors (Figure 2). The main microorganisms that cause AOM are bacteria and viruses, and their main route of infection is the eustachian tube. Viruses that cause OM include respiratory syncytial virus, rhinovirus, influenza virus, adenovirus, rhinovirus, parainfluenza virus, enterovirus, cytomegalovirus, mumps virus, measles virus, and herpes simplex virus. Of these, rhinovirus and adenovirus are the most frequently detected in upper respiratory tract infections, whereas the detection rate of rhinovirus in OM is lower than that of other viruses. Prior to introduction of the pneumococcal vaccine in 2000, *Streptococcus pneumoniae* was the most frequent bacterial cause of OM, followed by *Haemophilus influenzae* and *Moraxella catarrhalis*. In contrast, Group A ß–hemolytic streptococcus, α-streptococcus, *Staphylococcus aureus*, and *Pseudomonas aeruginosa* were rarely found in OM. However, after introduction of the pneumococcal vaccine, the relative *S. pneumoniae* detection rate decreased, whereas the proportion of cases positive for *H. influenzae* or *M. catarrhalis* increased. 

Acute inflammation of the middle ear results in hyperplasia and pathological transformation of the middle ear mucosa. Hyperplasia of the middle ear mucosa and invasion of various inflammatory cells into the mucosa are largely reversible, such that the mucosa undergoes de-differentiation and returns to its normal shape after the irritation associated with OM is removed. However, if pathological conditions such as middle ear mucosal hyperplasia, effusion from the hyperproliferative reaction, atelectasis, adhesions, tympanosclerosis, and cholesteatoma repeatedly occur and become chronic, irreversible structural changes in the middle ear cavity occur. Therefore, most cases of AOM resolve without sequelae, but in some cases, they progress in the form of recurrent OM, OME, or COM [4,5,6].

As with AOM, OME is mainly caused by microbial infection (i.e., bacteria or viruses). In the case of virally induced OME, it has been reported that rhinovirus is detected most often in the nasopharynx as well as the effusion. Viruses that cause upper respiratory tract infections are usually associated with secondary bacterial infections. For example, influenza A acts together with *S. pneumoniae* and respiratory syncytial virus acts together with *H. influenza* to cause infection. Representative bacteria identified in bacterial cultures of effusion include *S. pneumoniae*, *H. influenzae*, and *M. catarrhalis*. The negative detection rate in bacterial culture tests for OME is 40–60%. The causes of this high negative detection rate include slow or poorly growing bacteria, intracellular substances derived from viruses or anaerobes, and biofilms. Tests of bacterial cultures according to the viscosity of the effusion identified numerous bacteria in the mucous and purulent effusion, and showed that both recurrence rate and persistence rate increase as the viscosity of the effusion increases [4,7,8,9,10]. 

In COM, the most common aerobic bacteria is *P. aeruginosa*, and the most common anaerobic bacteria are Bacteroides species. Combined infection with aerobic bacteria and anaerobes is the main cause of ~50% of infections in COM. There are no differences in the composition of bacterial cultures between COM with cholesteatoma and COM without cholesteatoma. However, it is possible that the low frequency of reported bacterial species is attributable to biofilm formation [11]. COM without cholesteatoma causes changes in mucous membranes, submucosal tissue, and surrounding bone tissue as a result of a persistent inflammatory response in the middle ear cavity or mastoid air cells. Important pathological findings of COM include granuloma formation, bone changes, tympanosclerosis, cholesterol granuloma, cholesteatoma, and fibrosis. These pathologic findings appear differentially depending on whether the disease is in an active or inactive state. COM with cholesteatoma is a disease in which keratinized squamous epithelium invades into the mucosal membrane, accumulates keratin, and destroys the surrounding bone tissue. Various enzymes, including collagenase, acid phosphatase, and acid protease, secreted from granulation tissue produced by the inflammatory reaction as well as the pressure effect of the cholesteatoma itself are involved in bone destruction. It was recently shown that various types of cytokines secreted from inflammatory cells are also involved. Therefore, bone destruction occurs through activation of osteoclasts via these various mechanisms [12,13].

Otitis media is a common disease that requires surgical treatment if it does not improve with medical treatment. Improper treatment can lead to serious sequelae, including hearing loss, tympanic membrane perforation, tinnitus, ear fullness, facial nerve paralysis, mastoiditis, labyrinthitis, petrositis, postauricular abscess, Bezold’s abscess, zygomatic abscess, meningitis, brain abscess, extradural abscess, subdural abscess, otitic abscess, or otitic hydrocephalus. Therefore, there is an urgent need for methods to prevent the occurrence of and completely treat otitis media. This requires a better understanding of the development of otitis media and the defense mechanisms provided by the immune system in the middle ear cavity [14]. 

## 2. Innate Immunity in Otitis Media

OM commonly affects infants and children with an underdeveloped eustachian tube (E-tube). Importantly, in cases where a proper immune system is lacking or the immune reaction towards OM is dysfunctional, inflammation may become persistent and OM might progress to more severe diseases. Therefore, the middle ear immune system, particularly the innate immune system, plays an important role in the prompt and full recovery of OM. Notably in this context, the innate immune system is involved in maintaining physiochemical barriers through secretion of epithelial-barrier-related substances from the epithelium and effector proteins and cytokines from effector cells.

***Epithelial barriers***. The epithelial barrier is a physical barrier present in the normal epithelium of the middle ear that protects the epithelial cell layer in host tissue from microorganisms in the external environment. Epithelial cells, defensins, cathelicidins, and lymphocytes contribute to the epithelial barrier. The epithelial cells in the middle ear mucosa possess a chemical defense system that includes glycoproteins, surfactant, defensin, interferon, lactoferrin, defensive substances such as nitric oxide, and antibodies involved in adaptive immune responses [15]. 

***Surfactant***. Surfactant, which consists of phospholipid and protein, reduces the pressure required to open E-tubes by reducing surface tension. The opening pressure of the E-tube increases as the surface tension increases and the caliber of the E-tube decreases. Therefore, a lack of surfactant can lead to deterioration of dysfunctional E-tubes because of the impaired ventilation caused by the increased opening pressure. Among surfactant proteins (SPs), SP-D enhances bacterial elimination by increasing bacterial aggregation and phagocytosis. SP-D regulates innate immunity and inflammatory reactions in a mouse model of middle-ear OM induced by non-typeable *Haemophilus influenzae* (NTHi), and improves disease outcome by regulating NF-κB– and NLRP3-dependent activation of the inflammasome. The lack of SP-D increases inflammatory reactions to NTHi-induced ME infection and delays the resolution of OM compared with that in a WT mouse [16,17]. 

***Defensin***. Mucosal cells in the middle ear secrete various β-*defensins* (BDs) that act against different types of microorganisms. BD2, -3, and -4 are increased in the E-tube mucosa of OM model mice but not that of normal controls. In humans, it was reported that BD2 in the E-tube mucosa increases in response to NTHi-induced OM and cytokines such as interleukin-1α. BD3, which can be suppressed by biofilm, has an important role in the elimination of NTHi and is involved in recovery of OM [18,19].

***Interferons***. Both human and animal studies of OM have demonstrated the involvement of interferons in the pathogenesis of bacterial infection-triggered OM, showing that interferons are present in effusions from bacterial OM regardless of the microorganism strain [20]. In a previous study investigating interferon-γ levels in otorrhea of pediatric patients with acute suppurative OM, the mean level of interferon-γ secreted from the middle ear was 73.1 ± 9.5 pg/mL in patients with persistent and progressive inflammation to chronic suppurative OM, a significant increase compared with that in patients with improved acute suppurative otitis media (27.2 ± 8.8 pg/mL; *p* < 0.05). Moreover, discharge characteristics were altered in patients with mucoid, who showed a greater increase in measured values of interferon-γ (74.3 ± 19.1 pg/mL) compared with that in patients with other types of otorrhea (43.5 ± 15.6 pg/mL; *p* < 0.05). Moreover, interferon-γ concentrations were found to be significantly inversely correlation with IgG, IgE, and IgA concentrations (*p* < 0.05). These observations collectively suggest that increased concentrations of interferon-γ in middle ear secretions accelerate the progression to chronic suppurative OM [21]. 

***Lysozyme and lactoferrin.*** Immunohistochemical staining of E-tubes in mice has consistently revealed the presence of lysozymes in epithelial cells in the E-tube mucosa secreted by various kinds of mucous or serous secretory cells in the subepithelial gland. In contrast, lactoferrin is barely detectable in epithelial cells in the E-tube mucosa, but is found in serous secretory cells in the subepithelial gland. This lysozyme and lactoferrin distribution pattern is a species-specific characteristic that contributes to antibacterial defense reactions in the middle ear and E-tube. The amounts of lysozyme and lactoferrin are dynamically variable, dramatically increasing in the case of middle-ear infection and playing an important role in OM defense mechanisms and pathogenesis [22].

***Mucin***. Mucin is secreted by the nasal cavity or bronchi, and in the healthy state, protects and lubricates the mucosal layer. Expression of diverse mucus genes has been detected in the middle ear mucosa. Among the products of these genes, MUC1, MUC2, MUC5AC, MUC5B, MUC7, and MUC8 may have key roles in the pathogenesis of OM. In general, excessive mucus secretion is associated with inflammatory reactions and contributes to the etiology of OM [23]. 

***Aquaporins***. Members of the aquaporin (AQP) family of water channel proteins establish the biological hydration equilibrium and thus play essential roles in maintaining epithelial function in the middle-ear mucosa by ensuring proper hydration. To date, seven aquaporin genes—*AQP1, -3, -4, -5, -7, -8,* and *-9*—have been identified in mucosal cultures of middle-ear effusions. Although the precise role of aquaporins in the E-tube and middle ear have yet to be elucidated, these proteins are presumed to be involved in the production and excretion of middle-ear effusions, as exemplified by the observation that dysfunction of aquaporin regulation of periciliary fluid in the mucous membrane layer contributes to middle-ear effusion [24]. Taken together, 15 studies of mouse, rat, guinea, pig model and human studies indicate diverse patterns of AQP expression according to species and tissue type. Furthermore, the expression level of AQP was shown to depend on the presence or absence of inflammation, and was especially increased in patients with OM [8]. However, AQP4 and -6 mRNA expression levels were significantly decreased in patients with COM accompanied by otorrhea compared with COM patients without otorrhea. Therefore, either an increase or decrease in AQP expression level might have negative effects on fluid balance in the middle ear [24,25].

***Neutrophils***. Formation of neutrophil-driven neutrophil extracellular traps (NETs) has been reported in NTHi-mediated acute OM [26]. In this study using a chinchilla infection model, NETs were associated with high bacterial concentrations in middle-ear mucosa and effusions. 

***Macrophages***. Macrophages play an important role in innate immunity and are detected as a major cell component in human middle-ear effusions in OM [27].

***Dendritic cells***. Numerous dendritic cells are found in normal tympanic membranes, but the number of dendritic cells was shown to be increased in CSOM tympanic membranes compared with normal tympanic membranes. However, the role of dendritic cells in the middle ear and OM remains unclear and will require investigation [28].

***Natural killer (NK) cells***. The number of NK cells is increased in plasma of pediatric CSOM patients, indicating that these lymphocytes might be important in middle-ear infections [29].

***Complement***. Several members of the complement family—water-soluble molecules found in extracellular fluid and plasma that recognize pathogen-associated molecular patterns—act as immune-triggering molecules. The fact that expression of complement component 3a (C3a) is high in middle-ear effusions in recurrent OM suggests that complement components might reinforce inflammatory reactions in patients with OM. Furthermore, in an animal study employing an influenza A virus-induced pneumococcus OM model, C1qa-depleted mice exhibited delayed recovery from infection and showed severe damage in E-tubes and middle-ear mucosa, indicating that both classical and alternative complement pathways are important in immune reactions of the middle ear [30,31].

## 3. NOD-like Receptors (NLRs) as Pattern-Recognition Receptors (PRRs)

The most reliable way for a host organism to protect itself from a pathogen is to accurately recognize the presence of the pathogen and appropriately activate the immune response to neutralize it. Pathogen recognition is achieved by recognition of non-self molecules specific to the pathogen by receptors of the host organism. Substances that exist only in pathogens are currently called pathogen-associated molecular patterns (PAMPs). However, since immune responses of the host organism are regulated not only by pathogens but also by microorganisms that form a symbiosis/commensalism, microbe-associated molecular pattern (MAMP), a term with a wider applicable range, has come into frequent usage. The host’s immune responses are also induced by the recognition of self molecules released owing to host cell damage, which are called damage-associated molecular patterns (DAMPs). The host organism recognizes MAMPs or DAMPs through cellular receptors called pattern-recognition receptors (PRRs) [32,33,34]. The types of PRR include Toll-like receptors (TLRs), CLRs, nucleotide-binding oligomerization domain (NOD)-like receptors (NLRs), and retinoic acid-inducible gene receptors (RIG-Is). 

NLR consists of an N-terminal effector domain, a central NOD domain, and a common molecular organization called C-terminal leucine-rich repeats (LRRs). NLRs have been classified into several subclasses, NLRA, NLRB, NLRC and NLRP, based on the combination of N-terminal effector domains, including the transactivator domain (AD), baculovirus inhibitor repeats (BIRs), caspase recruitment domain (CARD), and pyrin domain (PYD). The central NOD domain possesses ATPase activity and is involved in NLR oligomerization [35,36]. NLRs are PRRs expressed in the cytosol and are composed of an LRR domain that recognizes a ligand, a NACHT domain for oligomerization of LRR, and a CARD or pyrin domain that directly binds to signaling molecules and transmits a signal. As a cytoplasmic receptor with a pyrin domain, 14 members of the NALP family and five types of NOD, CIITA, IPAF, and NAIP have been identified as a cytoplasmic receptor with a CARD [37]. NLRs are normally in a folded and repressed autorepressed form, but are activated by structural changes following binding to PAMP. When bacteria enter the cell through phagocytosis, peptidoglycans (PGNs), a component of the bacterial cell wall, are decomposed and the resulting MDP activates the repressed NOD and NALP3. Just as TLR transmits a signal through the interactions of MyD88 with the TIR domain, NOD activates NF-kB through RIP2 and CARD-CARD interactions, and NALP3 activates NF-kB through interactions with ASC and PYD-PYD interactions, thereby generating IL-1β. CIITA, the only member of the NLRA subgroup, induces the expression of MHC classes I and II. Although CIITA does not have a DNA-binding domain, it is involved in transcription by inducing the transcription machinery. NLRB in humans contains one NAIP, which binds to the Needle protein of the type III secretion system (T3SS). NAIP is an anti-apoptotic protein that inhibits caspases activated by MAPK. NLRC includes the most well-known NOD1 and NOD2, which recognize γ-d-glutamyl-meso-diaminopimelic acid (iE-DAP) and MDP, respectively, through their LRR domains. Ligand binding induces the release of an auto-inhibitory conformation, followed by oligomerization of NOD1/2, and recruitment of receptor-interacting protein kinases 2 (RIP2) through CARD-CARD interactions. Finally, NF-Kb and MAPKs are activated, resulting in antimicrobial responses. Typically, NOD-1 and NOD2 are expressed in the skin. NOD1 expression in human keratinocytes stimulates the expression of IFN-r and IL-8 in response to *Pseudomonas aeruginosa* [38]. NLRP3 is one of the best characterized molecules in the NLRP subgroup and is involved in inflammasome activation. NLRP3 activated by PAMPs and DAMPs recruits adapter apoptosis-associated speck-like protein containing a CARD (ASC) or pro-caspase-1. NLRP3 has been associated with several human diseases. The NLRP3 inflammasome is activated through a two-signal mechanism. The first signal is initiated by TLRs, IL-1 receptors, or tumor necrosis factor receptors (TNFRs), which stimulate the transcription factor NF-kB to produce IL-1β and IL-18 precursors. The second signal is induced by PAMPs and DAMPs, which convert pro-caspase-1 to caspase by inducing the oligomerization of NLRP3. The resulting activated caspase-1 induces cell lysis or pyroptosis [39]. In particular, NLRP3 is involved in the pathogenesis or progression of kidney disease. Ischemic reperfusion kidney injury induces cell damage or death by secreting DAMPs, which are also involved in rhabdomyolysis-associated acute kidney injury, as well as in lupus nephritis and systemic lupus erythematosus [39]. In addition, hypomethylation of NOD-1 and NOD-2, resulting in their aberrant expression, has been associated with chronic inflammatory diseases such as inflammatory bowel disease and rheumatoid arthritis. Recent studies have addressed the pharmacological mechanisms targeting the epigenetic modification of NLRs involved in the pathophysiology of these diseases [40]. NLRs act as intracellular sensors for bacterial infection; among the 23 known subtypes, NOD1 and NOD2 play important roles in the pathogenesis of OM. TLRs recognize pathogens on the cell surface or on the endosome/lysosome membrane and thus cannot recognize pathogens in the cytosol. Therefore, NLRs act as cytoplasmic receptors that recognize pathogens in the cytosol that have evaded recognition by TLRs. NLRs are normally in a folded, autorepressed form, but are activated by the binding of PAMPs, which induce a conformational change in the protein. Diseases associated with NLRs include OM, Crohn’s disease, Blau syndrome, and auto-inflammatory syndrome [41,42].

## 4. NLRs and OM

Nucleotide-binding oligomerization domain-like receptors, or NOD-like receptors (NLRs), are divided into four subfamilies based on the type of N-terminal domain:NLRA (A for acidic transactivating domain): CIITANLRB (B for BIRs): NAIPNLRC (C for CARD): NOD1, NOD2, NLRC3, NLRC4, NLRC5NLRP (P for PYD): NLRP1, NLRP2, NLRP3, NLRP4, NLRP5, NLRP6, NLRP7, NLPR8, NLRP9, NLRP10, NLRP11, NLRP12, NLRP13, NLRP14

This study reviewed three substances associated with otitis media: NOD1, NOD 2, and NLR family pyrin domain containing 3 (NLRRP3). (Table 1)

### 4.1. NLR1 and NLR2

In a study of 46 OME children who required ventilation tube insertion [40], NOD1 and NOD2 mRNA expression in middle-ear effusions collected during surgery was measured using quantitative polymerase chain reaction (qPCR). Expression levels of mRNA for these NLRs were compared according to the presence or absence of bacteria, characteristics of the middle-ear effusion, and reoperation on inserted ventilation tubes. Bacterial culture test showed a detection rate of 67.4%, with detected bacteria including coagulase-negative staphylococcus (CNS), methicillin-resistant *S. aureus* (MRSA), methicillin-sensitive *S. aureus* (MSSA), *Streptococcus pneumonia*, *Micrococcus* spp., *P. aeruginosa*, *Streptococcus viridians*, and *Acinetobacter lwoffii*. Middle-ear effusions were characterized as mucoid (36.9% of cases), serous (34.7%), mucopurulent (21.7%), and purulent (6.5%). Expression of NOD1 and NOD2 mRNA was detected in all middle-ear effusions collected from OME patients. However, there was no significant difference in the expression level of PRRs according to the characteristics of the effusion, the presence or absence of bacteria, or the frequency of ventilation tube insertion, indicating that the expression of NLRs was not affected by these parameters. However, expression of NLRs was linked to the pathogenesis of OME [43]. A separate study that investigated 95 pediatric OME patients aged 0 to 10 years who underwent ventilatory tube insertion [29] found differences in the expression of NLRs according to age. In this study, patients were divided into four age groups: 0–2 years, 2–4 years, 4–7 years and 7 years or older, and expression levels of NLR mRNAs were compared according to effusion characteristics, the presence of bacteria, and the presence or absence of comorbid diseases. In bacterial culture tests, the bacterial detection rate was 32.6%, and the detected bacteria included CNS, *H. influenzae*, *S. pneumoniae*, MRSA, *P. aeruginosa*, *Corynebacterium suppurative*, *S. aureus*, *S. viridians*, and *Acinetobacter baumannii*. Cases of middle-ear effusion were characterized as 70.5% mucoid, 22.1% serous, and 7.4% purulent. Expression levels of NOD1 and NOD2 mRNAs in middle-ear effusions were significantly lower in patients aged 2–4 and 4–7 years compared with those aged 0–2 and >7 years (*p* < 0.05). In particular, the expression level of NOD1 mRNA was significantly lower in culture-positive patients than in culture-negative patients (*p* < 0.05).

Significant changes in the expression of NLRs have also been reported in recurrent OME [42,43]. In this study, expression levels of NOD1 and NOD2 mRNA were compared between 27 otitis-prone patients (i.e., those with a history of OM more than three times in 6 months or four or more times in 1 year) and a non-otitis-prone group of 39 patients (i.e., those with a history of OM 1–2 times every 6 months or three times a year or less). The expression of NOD1 mRNA was lower in the otitis-prone group than in the non-otitis-prone group (*p* < 0.05), suggesting that the decrease in the expression of NLR mRNAs is related to the recurrence of OME [44].

Obesity is also a cause of OM in children. Obesity, a known risk factor for other diseases, has become a growing worldwide problem. An increase in childhood obesity, in particular, is a worldwide trend; currently, 20% of all children in the United States are obese. In 1995, the World Health Organization (WHO) included childhood obesity in their classification of international diseases and recognized it as a disease that should be actively managed and treated. In this context, a study addressed the association of NRL expression in OM with obesity [45], comparing the expression of NOD1, NOD2 and the levels of various cytokines and three types of immunoglobulins in middle-ear effusions collected during ventilation tube insertion. In this study, 219 OME children were divided into an obese group and a non-obese group. NOD2 expression was elevated in association with a significant reduction in IL-6, IL-12, and TNF-α mRNA in the obese group compared with the non-obese group, suggesting diminished NOD2-mediated expression of these factors (*p* < 0.05). However, there was no difference in IgG, IgA, or IgM in middle-ear effusions between the two groups (*p* > 0.05). Therefore, NLR-mediated cytokine mRNA expression, which is part of the innate immune response rather than the acquired immune response, is involved in the middle-ear cavity of OME in the context of childhood obesity.

Another study investigated the expression of NLRs in middle-ear tissues of humans and animals [46], comparing NOD2/RICK-dependent regulation of β-defensin 2 in middle-ear fluid of pediatric AOM patients, human middle ear epithelial cells, and C57BL/6 and NOD2^−/−^ (B6.129S1-Nod2tm1Flv) mice. In human middle ear epithelial cells, NOD2 silencing inhibited non-typeable *H. influenzae* (NTHi)-induced β-defensin 2 production, whereas NOD2 overexpression augmented it. An NOD2 deficiency has been shown to decrease inflammatory responses induced by intratympanic inoculation of NTHi and inhibit NTHi clearance in the middle ear. Thus, cytoplasmic release of internalized NTHi is involved in the pathogenesis of NTHi infection, and NOD2-mediated β-defensin 2 regulation plays a protective role against NTHi-induced OM [47].

In a study on the expression of NLRs in OM in animals [48,49], an NTHi inoculum was administered to the middle-ear cavity in C57BL/6 mice, and NOD1 and NOD2 signaling were analyzed over time. Compared with wild-type mice, NOD1- and NOD2-deficient mice were more susceptible to prolonged OM infection caused by NTHi. In NOD1-deficient mice, macrophage participation was decreased owing to a delayed neutrophil-induced inflammatory response and prolonged mucosal hyperplasia. In NOD2-deficient mice, bacterial clearance was delayed by an overall decrease in the number of white blood cells recruited to the middle ear. Thus, these studies show that NOD plays an important role in the pathogenesis of OM and recovery from it and reinforces the importance of innate immune signaling in the host defensive response.

### 4.2. NLRP3 (NLR Family Pyrin Domain Containing 3)

Two human studies and two studies using animal models have investigated the expression of NLRP3 in OM. In one study, expression of NLRP3 in middle-ear samples was compared between cochlear implant surgery patients with COM with cholesteatoma and those with COM and normal middle-ear mucosa through reverse transcription polymerase chain reaction (RT-PCR) analysis and immunohistochemical studies. NLRP3 protein was observed in infiltrated inflammatory cells of COM with cholesteatoma patients and COM patients, in association with significantly increased NLRP3 mRNA levels. These studies implicate the NLRP3 inflammasome in the pathogenesis of middle-ear disease and thus suggest that regulation of inflammasome-mediated inflammation could be a strategy for the treatment of COM. A second study [50] compared the expression of NOD2 and NALP3 in the chronically inflamed middle-ear mucosa of patients with chronic middle-ear disease (CMED) with that in the normal middle-ear mucosa. Immunohistochemical staining revealed NOD2 and NALP3 expression in the epithelium and lamina propria of the middle-ear mucosa in the CMED group, but not in the control group. NOD2 levels were also higher in the CMED group than in the control group. Collectively, these observations indicate that NOD2 and NALP3 are expressed in the mucosa of the middle-ear cavity of patients with COM, suggesting their involvement in the pathophysiology of AOM and CMED. In contrast, NOD1 was not detected [51]. In an animal model study, middle-ear fluid and mucosa of the temporal bone were collected after induction of OM by injecting lipopolysaccharide (LPS) into the middle ear of 6–10-week-old wild-type mice and male BALB/c mice lacking the macrophage migration inhibitory factor (*Mif*) gene (*Mif*^−/−^ mice). LPS induced an increase in NLRP3-positive infiltrating inflammatory cells in the middle ear of wild-type mice. Notably, LPS-induced increases in the numbers of inflammatory cells and NLRP3 immunostaining were attenuated in *Mif^−/−^* mice compared with wild-type mice, from which the authors concluded that regulation of the NLRP3 inflammasome and macrophage MIF are important targets in the treatment of OM. Finally, a second animal study [52] investigated inflammatory responses in BALB/c mice after injection of LPS or vehicle control (phosphate-buffered saline) into the middle ear, demonstrating an increase in IL-1β, NLRP3, ASC (adaptor apoptosis-associated speck-like protein containing a caspase activation and recruitment domain [CARD) and a pyrin domain [PYD)), and caspase-1. In addition, NLRP3, ASC, and caspase-1 proteins were observed in infiltrating inflammatory cells in the LPS-injected group. On the basis of their findings, the authors of this study suggested that activation of the NLRP3 inflammasome contributes to the pathogenesis of OM through regulation of the activity of IL-1β, concluding that IL-1β plays a central role in the inflammatory process of OM [53]. They further suggested that modulating the inflammasome-mediated inflammatory response by targeting this regulatory pathway could be a strategy for treating OM. (Table 2)

NLR activation of immune responses in the middle ear cavity can provide clues to the treatment of otitis media. Alternatively, the determination of differences in NLR expression levels or patterns in patients with AOM, OME, COM, and COM with cholesteatoma can provide indications for early-stage treatment, or as biomarkers for differentiating among these disease types or progression to complications.

## 5. Conclusions

Innate immune reactions involving epithelial cells, surfactant, defensin, interferon, lysozyme, lactoferrin, mucin, TLRs, CLRs, neutrophils, macrophages, dendritic cells, NK cells, and complement contribute to the pathogenesis of OM. NLRs are expressed in the fluid and middle ear mucosa of AOM, the effusion of OME, and the middle ear mucosa and inflamed tissue of COM. The expression of NLRs in OM in animals and humans shows different patterns depending on the presence or absence of bacteria, severity of inflammation, characteristics of the fluid, disease recurrence, tissue type, and repeated surgeries. Collectively, these observations support an important role for NLRs in the pathogenesis of OM.

## Figures and Tables

**Figure 1 ijms-23-02350-f001:**
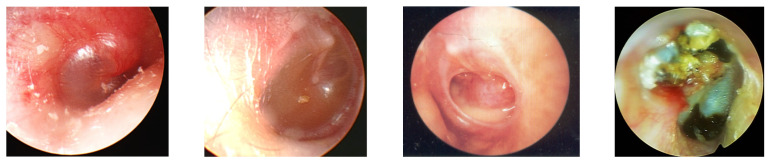
Tympanic membrane findings of AOM, OME, COM and CholeOM.

**Figure 2 ijms-23-02350-f002:**
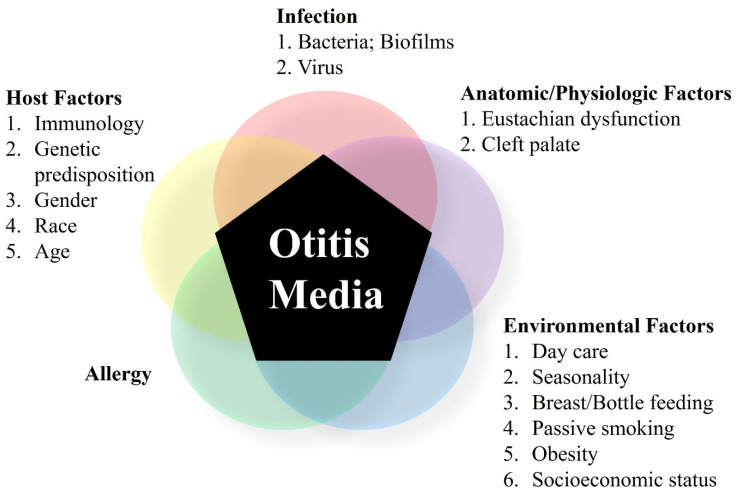
Various factors interact in the pathogenesis of otitis media.

**Figure 3 ijms-23-02350-f003:**
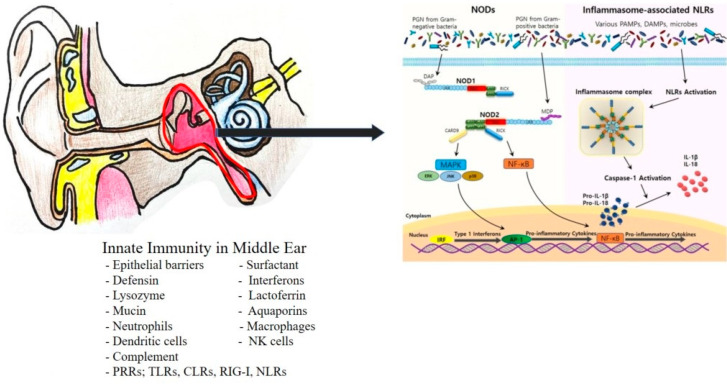
NLRs in otitis media. The middle ear is lined by epithelial cells, which can provide protection by secreting antimicrobial molecules, or through NLRs that provide defenses against intruding pathogens. Abbreviations: NOD—nucleotide-binding oligomerization domain; NLRP—nucleotide-binding oligomerization domain-like receptor protein; DAMP—damage-associated molecular pattern; PAMP—pathogen-associated molecular pattern.

**Table 1 ijms-23-02350-t001:** NOD-like receptors and their roles in protecting against pathogens.

NLR	Localization	Cell Types	Signaling Molecule/Adapter or Binder	Microorganism	Function
NOD1	Cytoplasm	Monocytes DCs Macrophages T and B lymphocytes Intestinal epithelium	RIP2 ATGL16	*Clostridium difficile* Gram-negative bacteria *Helicobacter pylori* *Listeria monocytogenes* *Salmonella typhimurium* *Shigella flexneri* *Aspergillus fumigatus*	NF-κB activation Autophagy
NOD2	Cytoplasm	Monocytes DCs Macrophages T and B lymphocytes Myeloid cells Bronchial epithelial cells	RIP2 CARD9	*A. fumigatus* *Citrobacter rodentium* *Escherichia coli* *Enterococcus faecalis* Gram-negative bacteria *Helicobacter hepaticus* *L. monocytogenes* *S. typhimurium* *Candida parapsilosis* Chitin	NF-κB activation Autophagy
NLRC3					Negative TLR regulator
NLRC4		Macrophages	ASC-NAIPS caspase-1	*C. rodentium* Legionella Pseudomonas Salmonella Shigella	Inflammasome formation
NLRC5				Virus	Transcription MHC class I-related genes
NLRX1					ROS production Autophagy Negative regulator of TLR MAVS-dependent signaling
NLRP1		Leukocytes Epithelial cells	ASC-caspase-1	*Bacillus anthracis*	Inflammasome formation
NLRP2					Inflammasome formation
NLRP3	Cytoplasm	Monocytes DCs Macrophages Neutrophils T and B lymphocytes Epithelial cells Myeloid cells	ASC Caspase-1	Bacteria *Listeria monocytogenes* *Neisseria gonorrhoeae* Staphylococcus *S. typhimurium* *A. fumigatus* *C. parapsilosis* *Cryptococcus neoformans* *Histoplasma capsulatum* *Malassezia* spp. *Paracoccidioides brasiliensis* *Sporothrix schenckii* Viral RNA	Inflammasome formation
NLRP4	Cytoplasm	DCs Macrophages	TBK1	*Candida albicans*	Autophagy Negative regulation of NF-κB
NLRP6				*E. coli* *L. monocytogenes* *S. typhimurium*	Inflammasome formation Negative TLR regulator
NLRP7					Inflammasome formation
NLRP10	Cytoplasm	DCs Macrophages Epithelial cells T lymphocytes	ASC Caspase-1	*C. albicans*	Negative regulation of NF-κB DC migration
NLRP12		Myeloid cells		*S. typhimurium* *Yersinia pestis*	Negative TLR regulator
NAIP					Inflammasome formation
NAIP2					Inflammasome formation
NAIP5					Inflammasome formation
AIM2			ASC-caspase-1	*Francisella tularensis*	
CIITA		Lymphocytes Endothelial cells			MHCII regulation

NOD—nucleotide-binding oligomerization domain; NLRP—nucleotide-binding oligomerization domain-like receptor protein; NLRC—NLR Family CARD Domain Containing.

**Table 2 ijms-23-02350-t002:** Studies assessing the association between OM and NOD-like receptors.

Author. Year [Reference]	Associated Diseases	Study Design	Species and/or Sample	Detection Method	Target Gene(s) or Pathway(s) Associated with NLRs	Results/Conclusion
Shin K, et al., 2020 [53]	OM	Animal study	Mice	ELISA; immunohistochemistry	NLRP3	IL-1β, NLRP3, ASC, and caspase-1 levels increased in LPS-treated wild-type mice; these increases were attenuated in LPS-treated *Mif*^−/−^ mice. /Macrophage MIF plays an important role in the production of IL-1β and the NLRP3 inflammasome.
Lee J, et al., 2019 [49]	OM	Animal study	Mice	Histology; macrophage phagocytosis and NTHi-killing assay; DNA microarrays	NOD1, NOD2	NOD1-KO mice appeared to have reduced macrophage enlistment with a delayed inflammatory response by neutrophils and prolonged mucosal hyperplasia, whereas NOD2-KO mice exhibited an overall reduction in the number of leukocytes recruited to the middle ear, leading to delayed bacterial clearance. /NODs play a role in the pathogenesis and recovery of OM, reinforcing the importance of innate immune signaling in the protective host response.
Kariva S, et al., 2018 [52]	AOM	Animal study	Mice	ELISA; immunohistochemistry.	NLRP3	Trans-tympanic injection of LPS significantly upregulated IL-1β, NLRP3, ASC, and caspase-1 in the middle ear compared with that in control mice and induced NLRP3 inflammasome components in the middle ear. /The NLRP3 inflammasome may play an important role in the pathogenesis of OM.
Kaur R, et al., 2016 [47]	AOM	Prospective study	Human: middle ear fluid	Quantitative PCR	NLR	Changes in innate gene regulation in AOM, measured in middle ear fluid, were similar whether caused by *S. pneumoniae* or NTHi. The innate immune response in otitis-sensitive children differed from that of children who were not otitis prone. /Defects in innate responses in the middle ear likely contribute to otitis proneness.
Kariya S, et al., 2016 [51]	COM, Chole OM	Prospective study	Human: Middle ear tissue samples	RT-PCR; immunohistochemistry.	NLRP3	NLRP3, ASC, and caspase-1 mRNA levels were significantly elevated in cholesteatoma and COM compared with that in normal controls. NLRP3, ASC, and caspase-1 protein were detected in infiltrating inflammatory cells in cholesteatoma and COM. /The NLRP3 inflammasome plays an important role in the pathogenesis of middle ear diseases. Modulation of inflammasome-mediated inflammation may be a novel therapeutic strategy for cholesteatoma and COM.
Kim SH, et al., 2015 [44]	OME	Prospective study	Human: Middle ear fluid	RT-PCR	NOD1 NOD2	TLR-2, TLR-9, NOD-1, NOD-2, IL-1, IL-6, and TNFα mRNA expression levels in effusion fluid were significantly higher in children aged 0–2 and >7 years (*p* < 0.05 each) than in those in 2–4 and 4–7-year groups. TLR-4, TLR-5, TLR-9, and NOD-1 mRNA expression levels were significantly lower in culture-positive than culture-negative patients (*p* < 0.05 each). /PRR and cytokine mRNA expression levels differ by age in children with OME.
Woo JI, et al., 2014 [46]	OM	Animal study	Mice	RT-PCR; ELISA; transmission electron microscopy; luciferase assay and gene silencing	NOD1 NOD2	NOD2 silencing inhibited NTHi-induced β-defensin 2 production in human middle ear epithelial cells, whereas NOD2 over-expression augmented it. NTHi-induced β-defensin 2 up-regulation was attenuated by cytochalasin D, an inhibitor of actin polymerization, and was enhanced by α-hemolysin, a pore-forming toxin. α-hemolysin-mediated enhancement of NTHi-induced β-defensin 2 up-regulation was blocked by silencing of NOD2. An NOD2 deficiency reduced inflammatory reactions in response to intratympanic inoculation of NTHi and inhibited NTHi clearance from the middle ear. /Cytoplasmic release of internalized NTHi is involved in the pathogenesis of NTHi infections, and NOD2-mediated β-defensin 2 regulation contributes to the protection against NTHi-induced OM.
Kim YJ, et al., 2014 [45]	OME	Prospective study	Human: Middle ear fluid	Quantitative PCR	NOD1 NOD2	NOD2-mediated expression of IL-6, IL-12, and TNF-α mRNA was significantly lower in obese than non-obese children (*p* < 0.05). /PRR-mediated cytokine mRNA expression is lower in obese than non-obese children with OME.
Lee SY, et al., 2011 [43]	OME	Prospective study	Human: Middle ear fluid	Quantitative PCR	NOD1 NOD2	PCR analyses showed that all effusion fluid samples collected from patients with OME expressed NOD1 and NOD2 mRNA. However, no differences in expression levels of PRRs in relation to characteristics of exudates, presence of bacteria, or frequencies of ventilation tube insertion were found. /Exudates of OME patients show expression of PRRs related to the innate immune response regardless of the characteristics of effusion fluid, presence of bacteria in exudates, or frequency of ventilation tube insertion.
Granath A, et al., 2011 [48]	CMED	Prospective study	Human: middle ear mucosa	Quantitative PCR; immunohistochemistry	NOD1 NOD2 NALP3	In the first report of its kind, clear immunohistochemical staining for NOD2 and NALP3 was detected in the epithelium and lamina propria of the middle ear mucosa. NOD1 was not detected, and no staining for NOD2 or NALP3 was observed in negative controls. The functions of NLRs in AOM and CMED remain a matter of speculation. /CMED might have an infectious origin regardless of clinical appearance, and inborn individual differences in the ability to express PRRs are part of its pathogenesis.
Kim MG, et al., 2010 [50]	Recurrent OME	Prospective study	Human: Middle ear fluid	Quantitative PCR; ELISA.	NOD1 NOD2	NOD1 mRNA levels were significantly lower in the otitis-prone than non-otitis-prone group. There was no correlation between immunoglobulin concentration and the expression of PRPs. /Decreased expression of PRRs is associated with increased susceptibility to OME.

NOD—nucleotide-binding oligomerization domain; NLRP3—nucleotide-binding oligomerization domain-like receptor protein 3; *Mif*^−/−^—macrophage migration inhibitory factor gene-deficient; ASC—apoptosis-associated speck-like protein containing a caspase recruitment domain and a pyrin domain; RT-PCR—*reverse transcription-polymerase chain reaction*; qRT-PCR—quantitative *reverse transcription*-polymerase chain reaction; ME—middle ear; MEF—middle ear fluid; WT—wild type; OM—otitis media; Chole OM—chronic otitis media with cholesteatoma; COM—chronic otitis media; CSOM—chronic suppurative otitis media; CMED—MEF—middle ear fluid; NTHi—nontypeable *H. influenza*; PRRs—Pattern recognition receptors.(Figure 3).

## Data Availability

Not applicable.

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
