# Peer review of "The Roles of NOD-like Receptors in Innate Immunity in Otitis Media"

_ijms, 2022, doi:10.3390/ijms23042350_

Round 1
Reviewer 1 Report
This manuscript introduces the involvement of inflammatory modules by NLRs in otitis media, particularly studies in animals and humans. It contains a brief background information about otitis media, the relevant innate immune systems, and NLRs. However, as review article, several modifications may be needed to make it easily accessible to the readers.
- It was difficult to figure out this manuscript’s scope, the reviewed contents. A brief explanation was found in Abstract, but it was too brief. What was included was found in the middle of the manuscript at the end of 3rd section (page 12) : To investigate the expression and role of NLRs in OM, we conducted ..... 11 articles satisfying search criteria were selected for comprehensive review. Reading the first 12 pages was not comfortable because it was not clear what was conducted in this manuscript. Maybe, it is necessary to locate these information in abstract or at the beginning of the main text.
- Similarly, no information of the manuscript’ writing structure was found. If the readers can see the list of contents at the beginning of the text, it would be much convenient.
- In section 4-2, NLRP3 appears first time without any explanation. Is that a name of inflammasome? The readers may need some explanation about NLRP3 with its full name and want to hear its relationship with the topic of the manuscript.
- The title of 4 looks premature. [Table 2] and reference numbers in a title are pretty embarrassing. By the way, The titles in section 4 look not well-prepared.
- Maybe, an additional brief text explaining what is going in the subsections under section 4 may help the readers follow the main finding of this manuscript.
- Ia that possible to prepare an additional concise figure summarize the conclusion under the conclusion section? If possible, it would make the manuscript beautiful.
Reviewer 2 Report
The manuscript titled Innate Immunity and Roles of NOD-like receptors in Otitis Media, is a report of a literature review with the following inclusion criteria:
"Literature databases were searched for studies on NLRs published in English or Korean and were included if they 1) were prospective investigational studies; and 2) included animal studies or human patients diagnosed with OM, AOM, OME, COM without cholesteatoma, or COM with cholesteatoma."
-> Revision point: Please specify in a methods subsection which literature databases were searched? With which program or web interface was the database searched and what was the search algorithm/code?
The 11 articles satisfying the search conditions, were not only discussed in detail, but also categorized in a beautiful list format.
The disease category of middle ear infections that includes a group of syndromes (acute ortitis media (AOM, ortitis media with effusion (OME) and chronic otitis media (COM)) represents a major human disease burden. "OME is very common, with 80% of children having had one or more episode of OME by 10 years of age." https://www.nature.com/articles/nrdp201663
The author's ORCID numbers and their respective contributions have been clearly stated. The reference list contains 56 articles, of which 10 articles are self references. The self reference rate is 18 percent.
On the topic of otitis media and pattern recognition receptors, the authors have previously published regularly in the following papers that were referenced in this manuscript (citations on Google Scholar):
28. Lee JH, Park DC, Oh IW, Kim YI, Kim JB, Yeo SG. C-type lectin receptors mRNA expression in patients with otitis media with effusion. Int J Pediatr Otorhinolaryngol 2013;77(11):1846-1851. (9 citations- 3 self citations)
27. Jung SY, Kim D, Park DC, Kim SS, Oh TI, Kang DW, Kim SH, Yeo SG. Toll-Like Receptors: Expression and Roles in Otitis Media. Int J Mol Sci 2021;22(15):7868. (0 citations, published last year)
38. Yeo SG, Won YS, Lee HY, Kim YI, Lee JW, Park DC. Increased expression of pattern recognition receptors and nitric oxide synthase in patients with endometriosis. Int J Med Sci 2013;10(9):1199-2008. (37 citations, 1 self citation )
51. Kim SH, Cha SH, Kim YI, Byun JY, Park MS, Yeo SG. Age dependent changes in pattern recognition receptor and cytokine mRNA expression in children with otitis media with effusion. Int J Pediatr Otorhinolaryngol 2015;79(2):229-
- (16 citations, 0 self citations)
53. Kim YJ, Cha SH, Lee HY, Lee SK, Chung HY, Yeo JH, Kim YI, Yeo SG Decreased pattern-recognition receptor-mediated cytokine mRNA expression in obese children with otitis media with effusion. Clin Exp Otorhinolaryngol 2014 ;7(1):7-12. (22 citations, 3 self citations)
54. Lee SY, Ryu EW, Kim JB, Yeo SG. Clinical approaches for understanding the expression levels of pattern recognition receptors in otitis media with effusion. Clin Exp Otorhinolaryngol.
2011;4(4):163-167. (18 citations, 1 self citation)
56. Kim MG, Park DC, Shim JS, Jung H, Park MS, Kim YI, Lee JW, Yeo SG. TLR-9, NOD-1, NOD-2, RIG-I and immunoglobulins in recurrent otitis media with effusion. Int J Pediatr Otorhinolaryngol 2010;74(12):1425-1429 (26 citations, 6 self citations)
Self citation reduces the scientific exchange of ideas, which is against the purpose of review articles. According to the systematic analysis of self citation rate, around 10 percent citation is normal [https://link.springer.com/article/10.1007/s11192-020-03417-5 ].
The conclusion of this formal analysis is that this manuscript's rate of self citation is too high. In general, the corresponding author can be identified as internationally referenced person.
->Revision point: Reduce the percentage of self citation to 10 percent.
The manuscript's main text is structured into 5 sections as listed in the following:
- Overview of otitis media
- Innate Immunity in Otitis Media
- NOD-like receptors (NLRs) as pattern-recognition receptors (PRRs)
- NLRs and OM
4-1. NLR1 and NLR2
4-2. NLRP3
- Conclusions
In the following each section is addressed separately.
Regarding section 1. Overview of otitis media:
It contains two high quality Figures. Fig. 1 shows images of human middle ear infections of different kind and at different stages. Fig. 2 gives a good schematic overview of contributing factors that are associated with Otitis Media (OM). This section clearly distinguishes many different kinds of Otitis Media and details the different pathogens that were found in OM patients. The discussion of why 40-60 percent of bacterial culture tests from patients is negative is very important.
Regarding section 2. Innate Immunity in Otitis Media:
This section summarizes many important factors of the inflammatory environment in the middle ear. A pragraph is written for each of the following subjects: Epithelial barriers, Surfactant, Defensin, Interferons, Lysozyme & lactoferrin, Mucin, Aquaporins, Toll-like receptors, C-type lectin receptors, Neutrophils, Macrophages, Dendritic cells, Natural killer cells. This section of around 1700 words mainly functions to elaborate on the terminology, but distracts from the topic of the review given in the title: "Innate Immunity and Roles of NOD-Like Receptors in Otitis Media". If the review is on the Innate Immunity's Role in Otitis Media, which is opposition to the author's statement: "Literature databases were searched for studies on NLRs published in English or Korean and were included if they 1) were prospective investigational studies; and 2) included animal studies or human patients diagnosed with OM, AOM, OME, COM without cholesteatoma, or COM with cholesteatoma." In addition, the authors have already published separate reviews on the some of these topics.
As an example, the paragraph on Toll like receptors contains one citation to one of the author's previous reviews that focuses fully on TLRs and Otitis Medians [27. Jung SY, Kim D, Park DC, Kim SS, Oh TI, Kang DW, Kim SH, Yeo SG. Toll-Like Receptors: Expression and Roles in Otitis Media. Int J Mol Sci 2021;22(15):7868. (0 citations, published last year)]. There is no novelty.
Otitis media clearly is a hot topic, because the pubmed database search for the terms "otitis+media" finds 913 publications in 2021 alone. There were 1880 paper published on "nod-like+receptor" in 2021. With an average publication date of around 2008, this manuscript's references are not recent enough. The distribution of the publication dates is shown here:
-> Revision point: Either discuss most recent data on Otitis media, or cut the whole section and write a section about how what is the newest information about how NLRs creates an inflammatory environment. Include most recently published papers.
Regarding section 3. NOD-like receptors (NLRs) as pattern-recognition receptors (PRRs):
Pattern recognition receptors are key mechanistic factors for the immune reaction in middle ear infections. This manuscript focuses on the nucleotide-binding oligomerization domain (NOD)-like receptor. The focus is unfortunately placed on explaining the general concept of pattern recognition receptors, but fails to detail the molecular mechanism of NLRs.
The paragraph starting with "To investigate the expression and role …" needs to be cut our and placed into a methods section of this manuscript, as referred to earlier.
-> Revision point: Please cut down on previously published and discussed pattern recognition receptors and focus on the NLR. Following questions should be addressed: What family members of NLRs are of importance (only NOD1 and NOD2 are mentioned, please extend it)? How do they structurally/mechanistically differ from related protein families? How is NLR expression regulated? What are the post-translational modifications that regulate the proteins stability? What ligands have been reported for these receptors? What is the molecular mechanism of NLR activation? Are there other receptor families that detect the same type of ligand? What are the interaction partners of NLR inside the cell? Which cell types express these NLR isoforms [suggested resource: https://www.proteinatlas.org/ ]? Can the different NLRs differentiate between different pathogens?
Regarding section 4. NLR and OM: This section is the strongest section of this manuscript. It focuses on the "expression" of NLRs. Expression levels could mean mRNA or Protein. Sometimes "mRNA" is mentioned sometimes not. What is actually meant was not clearly indicated. The different bacteria species that were found in OME are very comprehensively listed. The correlation between NLR mRNA expression and specific disease characteristics are well described for 5 studies in the subsection "NLR1 and NLR2".
Regarding section 5. Conclusion: The conclusion section is very short and could be extended by discussion of what future research is suggested by the authors. The repeated listing of innate immune factors seems misplaced, because this was already assumed through section 2. This section mainly states that NLRs are expressed in AOM and thereby repeats the message from section 4.
->Revision point: Please write your expert opinion on what clinical trials or experiments should be undertaken to further the fields understanding of NLR's role in OM. Add a discussion about how medical professionals could increase their OM diagnostics. Especially address the question, if NLR expression could be seen as a biomarker to distinguish different types of otitis media.
Herewith, I recommend the re-reviewing of a revised manuscript.
Round 2
Reviewer 2 Report
The manuscript has been improved, but not enough to be published yet. The speed of the revision is reflected by the incoherence of changes in the structure of the manuscript. Good writing needs time. Please take your time, address all the reviewing comments appropriately and create a publication ready manuscript. The cover letter indicates some changes in the manuscript that were not present in the PDF file given to the reviewer and named: ijms-1461535-peer-review-v2 .
Following revision points were not appropriately addressed:
-> Revision point: Please cut down on previously published and discussed pattern recognition receptors and focus on the NLR. Following questions should be addressed: What family members of NLRs are of importance (only NOD1 and NOD2 are mentioned, please extend it)? How do they structurally/mechanistically differ from related protein families? How is NLR expression regulated? What are the post-translational modifications that regulate the proteins stability? What ligands have been reported for these receptors? What is the molecular mechanism of NLR activation? Are there other receptor families that detect the same type of ligand? What are the interaction partners of NLR inside the cell? Which cell types express these NLR isoforms [suggested resource: https://www.proteinatlas.org/ ]? Can the different NLRs differentiate between different pathogens?
While some questions were answered, the major revision point of cutting down text sections of unfocused parts was not appropriately done. It is easy to see that this revision point was not followed, just look at the section titled: "Toll-like receptors (TLRs)" and "C-type lectin receptors (CLRs)"
Please make sure to clearly indicate t the sections removed, so that the reviewer can more easily see what has been removed.
->Revision point: Please write your expert opinion on what clinical trials or experiments should be undertaken to further the fields understanding of NLR's role in OM. Add a discussion about how medical professionals could increase their OM diagnostics. Especially address the question, if NLR expression could be seen as a biomarker to distinguish different types of otitis media.
The following sections were added, which would fit better into the introduction of this manuscript:
"Otitis media is a common disease that requires surgical treatment if it does not improve with medical treatment. Improper treatment can lead to serious sequelae, including hearing loss, tympanic membrane perforation, tinnitus, ear fullness, facial nerve paralysis, mastoiditis, labyrinthitis, petrositis, postauricular abscess, Bezold’s abscess, zygomatic abscess, meningitis, brain abscess, extradural abscess, subdural abscess, otitic abscess, or otitic hydrocephalus. Therefore, there is an urgent need for
methods to prevent the occurrence of and completely treat otitis media. This requires a better understanding of the development of otitis media and the defense mechanisms provided by the immune system in the middle ear cavity."
The guiding principle of a conclusion could be rephrased by: What can you conclude from this manuscript's findings?
The second text part and figure that was added into the conclusion section belongs to the main text: "(Figure 3). NLR activation of immune responses in the middle ear cavity can provide clues to the treatment of otitis media. Alternatively, the determination of differences in NLR expression levels or patterns in patients with AOM, OME, COM, and COM with cholesteatoma can provide indications for early stage treatment, or as biomarkers for differentiating among these disease types or progression to complications."
